# INSTIs and NNRTIs Potently Inhibit HIV-1 Polypurine Tract Mutants in a Single Round Infection Assay

**DOI:** 10.3390/v13122501

**Published:** 2021-12-14

**Authors:** Steven J. Smith, Andrea Ferris, Xuezhi Zhao, Gary Pauly, Joel P. Schneider, Terrence R. Burke, Stephen H. Hughes

**Affiliations:** 1HIV Dynamics and Replication Program, Center for Cancer Research, National Cancer Institute, Frederick, MD 21702, USA; smithsj2@mail.nih.gov (S.J.S.); andrea.ferris@nih.gov (A.F.); 2Chemical Biology Laboratory, Center for Cancer Research, National Cancer Institute, Frederick, MD 21702, USA; xuezhi.zhao@nih.gov (X.Z.); pauly@ncifcrf.gov (G.P.); joel.schneider@nih.gov (J.P.S.); burkete@mail.nih.gov (T.R.B.J.)

**Keywords:** inhibition, integration, resistance, efficacy, infectivity

## Abstract

Integrase strand transfer inhibitors (INSTIs) are a class of antiretroviral compounds that prevent the insertion of a DNA copy of the viral genome into the host genome by targeting the viral enzyme integrase (IN). Dolutegravir (DTG) is a leading INSTI that is given, usually in combination with nucleoside reverse transcriptase inhibitors (NRTIs), to treat HIV-1 infections. The emergence of resistance to DTG and other leading INSTIs is rare. However, there are recent reports suggesting that drug resistance mutations can occur at positions outside the integrase gene either in the HIV-1 polypurine tract (PPT) or in the envelope gene (*env*). Here, we used single round infectivity assays to measure the antiviral potencies of several FDA-approved INSTIs and non-nucleoside reverse transcriptase inhibitors (NNRTIs) against a panel of HIV-1 PPT mutants. We also tested several of our promising INSTIs and NNRTIs in these assays. No measurable loss in potency was observed for either INSTIs or NNRTIs against the HIV-1 PPT mutants. This suggests that HIV-1 PPT mutants are not able, by themselves, to confer resistance to INSTIs or NNRTIs.

## 1. Introduction

Integrase strand transfer inhibitors (INSTIs), when used in combination with nucleoside reverse transcriptase inhibitors (NRTIs), are the currently recommended treatment strategy for people living with HIV who are either treatment-naïve or treatment-experienced [1]. Dolutegravir (DTG) and bictegravir (BIC) are potent INSTIs that comprise, together with cabotegravir (CAB), the second generation INSTIs. BIC and DTG retain high potencies against resistant mutants that arise in response to the first generation INSTIs raltegravir (RAL) and elvitegravir (EVG) [2,3,4,5,6,7]. When compared to RAL and EVG, the resistance profiles of DTG and BIC are much more favorable [8,9]. Only a few DTG- and BIC-resistant mutants have been identified either in selection studies in vitro or in HIV-infected individuals who are treatment-experienced [2,3,10,11]. Most mutations that are selected by antiretroviral drugs arise in the region that encodes the target protein [12]. For example, most drug-resistant mutations that arise in response to INSTIs are in the region of the *pol* gene that encodes IN. Although DTG and BIC are much less apt to select for resistance than RAL and EVG, the following mutations in IN were selected by DTG in an in vitro experiment: H51Y, T66A/I, G118R, E138K, S153Y/F, and R263K [2,13,14,15]. In addition, the M50I, S153Y/F, R263K, and M50I/R263K mutations in IN were selected by BIC [3]. Resistance mutations have been selected in vitro in the presence of CAB primarily at position Q148R/K. Several additional mutations emerged when viruses with amino acid substitutions at position Q148K/R and an additional IN mutant, Q148H, were passaged in cells in culture [16,17,18]. The Q148R mutant in IN was also selected in two people living with HIV who were in clinical trials that included CAB [19].

A recent study, using DTG selection in vitro, identified mutations in the HIV-1 polypurine tract (PPT) [20]. The PPT mutations were reported to confer resistance to DTG. Similar mutations in the HIV-1 PPT were reported to have been selected in a HIV-1 treatment-experienced subject who failed a DTG monotherapy maintenance regimen [21]. It has also been recently reported that mutations in the HIV-1 envelope glycoprotein (Env) can confer resistance to both DTG and RAL in multicycle spreading infections in cultured cells [22,23]. These reports suggest that it may be easier for HIV to develop resistance to the most potent and broadly effective INSTIs by acquiring mutations outside of the IN-coding region. However, the mechanisms underlying resistance that involve mutations outside the region that encodes the target protein are not well-defined.

Recently, a phenotypic analysis was performed to determine the susceptibility of HIV-1 PPT mutants to INSTIs (RAL, EVG, DTG, and CAB) and non-nucleoside reverse transcriptase inhibitors (NNRTIs) [efavirenz, (EFV) and rilpivirine (RPV)] [24]. This study suggested that the FDA-approved INSTIs and NNRTIs retain efficacy against two HIV-1 PPT mutants and concluded that these PPT mutations do not confer resistance to either class of drugs. To help resolve these conflicting reports, we determined the EC_50_ values of all the FDA-approved INSTIs, for the leading FDA-approved NNRTIs, and for several of our potent INSTI leads (Figure 1) and NNRTI leads (Figure 2) against a panel of HIV-1 PPT mutants that have been reported to confer resistance in a single round infection assay. We show that all of the INSTIs and NNRTIs we tested retained their potency against the HIV-1 PPT mutants. Additionally, the HIV-1 PPT mutants replicated poorly in single-round infection assays when compared to WT HIV-1. Our results show that the HIV-1 PPT mutants do not confer resistance to either INSTIs or NNRTIs in a single-round infection assay.

## 2. Materials and Methods

Vector constructs. The pNLNgoMIVR-Emod Luc vector has been described [25]. To generate the mutations used in this study, a 1.7 Kb XhoI–NcoI digested fragment containing the PPT was excised from pNLNgoMIVR-Emod Luc and subcloned into pET-32a(+). The resulting plasmid served as the template for site-directed mutagenesis (QuickChange II XL kit, Agilent Technologies, Santa Clara, CA, USA) of the poly G tract (GGGGGG) using the following sense oligonucleotides with matching cognate antisense oligonucleotides (not shown) (Integrated DNA Technologies, Coralville, IA):
AGC (GGGAGC), 5′-CCACTTTTTAAAAGAAAAGGGAGCACTGGAAGGGCTAATTC-3′;AGT (GGGAGT), 5′-CCACTTTTTAAAAGAAAAGGGAGTACTGGAAGGGCTAATTC-3′;+G (GGGGAGT), 5′-CCACTTTTTAAAAGAAAAGGGGAGTACTGGAAGGGCTAATTC-3′;+G (GGGGAGT), 5′-CCACTTTTTAAAAGAAAAGGGGAGTACTGGAAGGGCTAATTC-3′;ΔG (_GCAGT), 5′-CCACTTTTTAAAAGAAAAGCAGTACTGGAAGGGCTAATTC-3′;AGTG (GGAGTG)’ 5′-CCACTTTTTAAAAGAAAAGGAGTGACTGGAAGGGCTAATTC-3′.

Sequences of the mutated inserts were verified by DNA sequencing, and the XhoI–NcoI fragment was excised from each mutated plasmid and inserted into identically digested pNLNgoMIVR-Emod Luc.

Cell-based assays. The EC_50_ values of the compounds were measured using either WT HIV-1 or the HIV-1 PPT mutants in single-round infectivity assays as described previously [26,27]. Briefly, a VSV-g-pseudotyped HIV vector was produced by transfections of 293 T cells with pNLNgoMIVR-EmodΔLUC and pHCMV-g (obtained from Dr. Jane Burns, University of California, San Diego, CA, USA) using the calcium phosphate method. Approximately 6 h after the calcium phosphate precipitate was added, the transfected cells were washed twice with phosphate-buffered saline (PBS) and incubated with fresh media for 48 h. The virus-containing supernatants were then harvested, clarified by low-speed centrifugation, filtered, and diluted for use in antiviral infection assays. On the day prior to the assay, HOS cells were seeded in a 96-well luminescence cell culture plate at a density of 4000 cells in 100 µL per well. On the day of the assay, cells were treated with compounds over a range of concentrations from 5 µM to 0.0001 µM using 11 serial dilutions. After a 3 h preincubation with the compounds, 100 µL of virus-stock was added. The virus was diluted to achieve a luciferase signal between 0.2 and 2.0 Relative Luciferase Units (RLUs). The cells were incubated at 37 °C for 48 h. Infectivity was measured by using the Steady-lite plus luminescence reporter gene assay system (PerkinElmer, Waltham, MA, USA). Luciferase activity was measured by adding 100 µL of Steady-lite plus buffer (PerkinElmer) to the cells, incubating at room temperature for 20 min, and measuring luminescence using a microplate reader. Antiviral activities were normalized to the infectivity in cells in the absence of target compounds. KaleidaGraph (Synergy Software, Reading, PA, USA) was used to perform non-linear regression analysis on the data. EC_50_ values were determined from the fit model. The luciferase activity of the WT virus was set to 100%, and the infectivity of the mutant viruses was measured as a percentage of WT.

INSTI and NNRTI synthesis. RAL, EVG, DTG, BIC, and CAB were acquired as described previously [4,28]. The preparation of compounds **4c**, **4d**, **4f**, **6b**, and **6v** was done by previously reported procedures [27,29,30]. EFV, RPV, and doravirine (DOR) were acquired as previously described [31,32]. The preparation of compounds **6**, **7**, **11**, **12**, **13**, **16**, and **27** has been reported [32,33]. 

## 3. Results

HIV-1 PPT mutants. Recent reports suggested that there are several different mutations in the HIV-1 PPT that can cause resistance to DTG [20,21]. An in vitro selection study identified several HIV-1 PPT mutations within the “poly G tract”, (GGGGGG) and a clinical study reported that a patient who suffered virological failure during DTG maintenance therapy had two mutations in the poly G tract, GGGAGC. Using the reported sequences of the mutant PPTs as guides, we made the following HIV-1 mutants that had changes in the poly G tract of the PPT (Figure 3): AGC (GGGAGC), AGT (GGGAGT), +G (GGGGAGT), ΔG (_GCAGT), and AGTG (GGAGTG).

Replication of the HIV-1 PPT mutants in a single-round infection assay. In general, INSTI-resistant mutants have a reduced ability to replicate when compared to WT HIV-1 [27,28,34]. To determine the degree to which the mutations in the HIV-1 PPT effect HIV-1 replication, we measured the replication of the HIV-1 PPT mutants and WT HIV-1 in single-round infection assays (Table 1).

Table 1 Replication of the HIV-1 PPT mutants using a single-round infection assay. Replication of the HIV-1 PPT mutants used in single-round infection assay. The abilities of the HIV-1 PPT mutants to replicate, compared with WT HIV-1, were measured in a single-round infection assay using vectors carrying the appropriate mutations. The relative luciferase activity for the WT HIV-1 virus in the absence of antiretroviral drugs was set to 100%, and infections by the HIV-1 vectors with the mutant PPTs (adjusted for amount of p24/Gag used in the assay) were measured and compared to WT. An asterisk indicates that the ΔG and AGTG did not replicate to a measurable extent in the assay.

All five of the HIV-1 PPT mutants were significantly less able to infect cells in a single-round assay relative to WT HIV-1. When compared to the replication of WT HIV-1 (100%), the AGC PPT mutant had a single round infectivity of 5.0%. The AGT and +G mutants had single-round infectivities of 7.0% and 8.0%, respectively. The additional PPT mutants we tested, ΔG and AGTG, did not infect cells to a measurable extent in our single-round assay. Our in vitro results suggest that the PPT mutants would probably replicate poorly in people living with HIV. However, Wei and Sluis-Cremer [35] reported that HIV carrying the mutant PPT we call AGC (which they call PPT2) replicated well in cultured cells. The virus/vectors are different, as are the assays; however, we cannot account for this apparent discrepancy.

Antiviral potencies of INSTIs against a panel of HIV-1 PPT mutants. Recent reports presented conflicting results on the susceptibility of HIV-1 PPT mutants to INSTIs [20,24]. To determine if there is a measurable loss in potency for INSTIs against the HIV-1 PPT mutants, the EC_50_ values were determined in single-round infection assays. We tested all five FDA-approved INSTIs, BIC, DTG, CAB, RAL, and EVG, and several of our most promising INSTIs: **4c**, **4c**, **4f**, **6p**, and **6v** against WT HIV-1 and the HIV-1 PPT mutants AGC, AGT, and +G (Figure 4, Appendix A).

All of the INSTIs we tested, except for 6v (195.0 ± 27.2 nM), potently inhibited the replication of WT HIV-1 (<4.0 nM). In agreement with the results of Wei and Sluis-Cremer [24], neither the FDA-approved INSTIs nor our compounds showed a loss of potency against the HIV-1 PPT mutants. When tested against the HIV-1 PPT mutant AGC, all of the INSTIs, except for 6v (85.4 ± 7.5 nM), had EC_50_ values below 3.0 nM. Similarly, for the AGT mutant, the EC_50_ values for BIC, DTG, EVG, **4d**, **4f**, and **6b** were all below 1.0 nM. CAB, RAL, and **4c** had potencies ranging between 2.0 to 4.0 nM for this HIV PPT mutant. Compound **6v** was the least potent (60.5 ± 3.8 nM) against AGC. The HIV-1 PPT mutant +G did not display a reduction in susceptibility against the INSTIs tested. All INSTIs had EC_50_ values below 1.0 nM against +G, except for RAL (2.1 ± 0.3 nM), **4c** (2.8 ± 0.5 nM) and **6v** (123.9 ± 3.3 nM). We saw no significant reduction in susceptibility to any of the HIV-1 PPT mutants we tested, using either the FDA-approved INSTIs or our compounds. The most consistent differences we saw were for **6v**. When compared to its EC_50_ value against WT (195.0 ± 27.2 nM), **6v** had a lower EC_50_ for all three HIV-1 PPT mutants, AGC, AGT and +G (85.4 ± 7.5 nM, 60.5 ± 3.8 nM, and 123.9 ± 3.3 nM, respectively). These differences suggest that the PPT mutants we tested may be slightly more susceptible to **6v**. However, the compound is not particularly potent, and the fold change in the EC_50_ is small (about 2-fold).

Comparison of antiviral potencies of NNRTIs against a panel of HIV-1 PPT mutants. It has been previously reported that RT-mediated plus strand DNA synthesis from the 3′PPT is inhibited by NNRTIs [35,36,37,38]. To show that NNRTIs can still effectively inhibit HIV-1 replication, and, by extension, reverse transcription of the HIV-1 PPT mutants, we measured the EC_50_ values of the FDA-approved NNRTIs EFV, RPV, and DOR and several of our NNRTIs (**6**, **7**, **11**, **12**, **13**, **16**, and **27**) against the HIV-1 PPT mutants AGC, AGT, and +G in a single-round infection assay (Figure 5, Appendix A).

We previously showed that the clinically relevant NNRTIs and several of our own NNRTIs exhibit subnanomolar EC_50_ values against WT HIV-1 [31,32,33]; the EC_50_ values measured in the assays performed for this report were similar. We saw no loss in potency when the FDA-approved NNRTIs and our promising NNRTIs were tested against the HIV-1 PPT mutants AGC, AGT, and +G. These data provide additional support for the conclusion that the HIV-1 PPT mutants do not confer significant resistance against either INSTIs or NNRTIs in single-round HIV-1 infection assays.

## 4. Conclusions

INSTIs have been used in the HIV-1 combination antiretroviral therapies that are currently recommended for either treatment-naïve or treatment-experienced individuals [1]. DTG and BIC potently inhibit WT HIV-1 and many of the signature RAL- and EVG-resistant mutants [2,3,4,5,6,7]. In addition, neither DTG nor BIC readily select new drug-resistant mutants. Resistance has been reported to arise against these INSTIs in vivo in treatment-experienced, INSTI-experienced individuals, who displayed virological failure in a RAL-containing regimen and were subsequently switched to a DTG-containing regimen [10,11]. However, there have been recent reports suggesting that there may be novel mechanisms of INSTI resistance that do not involve mutations in the region of *pol* that encodes IN. HIV-1 *env* mutants have been selected in vitro that broadly decrease the potencies of several types of anti-HIV drugs, including the INSTIs DTG and RAL, in multi-round in-vitro replication assays [22,23]. It has also been reported that mutations in the HIV-1 PPT were selected in vitro and that similar mutations were reported to have been selected in vivo in an individual who was failing a DTG maintenance monotherapy [20,21].

Here, we investigated whether the HIV-1 PPT mutants reduce the potencies of either INSTIs or NNRTIs in single-round infection assays. We measured EC_50_ values of the FDA-approved INSTIs and NNRTIs and several of our promising INSTIs and NNRTIs against a panel of HIV-1 PPT mutants. Neither the INSTIs nor the NNRTIs we tested showed a loss in potency against the HIV-1 PPT mutants, substantiating the results reported in a recent study [24]. In addition, when we tested the ability of the HIV-1 PPT mutants to replicate in single-round infection assays, all of the mutants we tested replicated poorly compared to WT HIV-1. Taken together, it seems likely that these HIV-1 PPT mutants do not provide significant levels of resistance to DTG or RAL.

These results are not surprising, given the nature of the IN strand transfer (ST) reaction. After the viral DNA is synthesized, a multimeric form of IN forms a complex with both ends of the newly synthesized linear viral DNA [39]. In the first step of the integration process called 3′-processing, IN removes a pair of nucleotides from each of the 3′-ends of the viral DNA [40,41]. In the second step, IN inserts the processed 3′ ends of the viral DNA into the host genome [42,43]. It is the second reaction, strand transfer, that is blocked by INSTIs.

The HIV-1 PPT, after being cleaved by the RNase H of RT, acts as the primer for second (positive) strand viral DNA synthesis [44]. Thus, the PPT plays a crucial role in reverse transcription, and mutations in the HIV-1 PPT can affect the specificity of the RNase H cleavage events [35]. This can, in turn, affect the sequence of the PPT primer. The sequence of the PPT primer defines where plus strand DNA synthesis is initiated, and, by extension, the sequences at the end of the upstream LTR in the linear viral DNA intermediate. IN may not be able to correctly process or integrate an abnormal linear viral DNA end. We previously showed that, if a viral DNA has one normal and one abnormal DNA end, the normal end can be integrated by IN, and the abnormal end can be inserted, with a reduced efficiency, by host enzymes [45]. The host-mediated integration event is abnormal and affects both the host and viral sequences at the junction. However, this type of abnormal viral/host integration event can lead to the creation of a provirus that can produce normal viral RNA. The published data strongly suggest that the host-mediated integration of the aberrant viral DNA end happens after the IN-mediated integration of the normal DNA end [46].

It is not clear how a PPT mutation would be able to protect the integration of the normal end of the linear viral DNA from an INSTI. The initial IN-mediated integration of the downstream end of the viral DNA should still be completely susceptible to inhibition by an INSTI. Even if the INSTI were to bind poorly to a complex of the aberrant viral DNA end bound to IN, IN would still have to be able to integrate the aberrant end reasonably efficiently. However, we know that the PPT mutants have a very low titer in the absence of an INSTI, which suggests that IN does not insert the mutant forms of the DNA made by the PPT mutants efficiently. The low titer of the PPT mutants was determined in the absence of an INSTI under conditions in which IN would be able to insert the normal end of the viral DNA efficiently. The titer of the virus would be further reduced if an INSTI were to block the integration of the normal end of the DNA. Thus, it is unclear how a PPT mutant could confer resistance to an INSTI.

Similar to INSTIs, we would not expect the HIV-1 PPT mutants to show a decrease in susceptibility to NNRTIs. NNRTIs inhibit DNA initiation for both the minus strand and plus strand during reverse transcription [38,47,48,49]. The minus strand transfer step and the RNase H cleavages that generate the PPT primer must occur before plus strand DNA synthesis is initiated. In addition, RT must make the entire minus strand DNA. NNRTIs should be able to inhibit the reverse transcription of the minus strand DNA in viruses that carry the HIV-1 PPT mutants. In addition, there are a number of mutations in HIV RT that can cause resistance to NNRTIs [50]. Changes in residues in and around the NNRTI binding pocket reduce the ability of RT to bind NNRTIs [32,33], making it less likely that NNRTIs would select for mutations outside the region of *pol* that encodes RT. Conversely, HIV has much more difficulty developing resistance to the second-generation INSTIs, particularly DTG and BIC [8].

Because it is difficult for HIV to develop resistance to the most potent second generation INSTIs, it is possible that the virus will develop mechanisms that do not involve changes in IN. There is good evidence that mutations in *env* can broadly reduce the susceptibility of HIV-1 to all antiviral drugs, including INSTIs, in a multi-round assay in vitro [22,23]. Although we think that all positions where resistance mutations have been reported to arise in the HIV-1 viral genome should be explored so that any novel mechanisms that could lead to treatment failure are fully explored, it does not appear that any of the reported PPT mutants confer resistance to INSTIs in single-round assays.

## Figures and Tables

**Figure 1 viruses-13-02501-f001:**
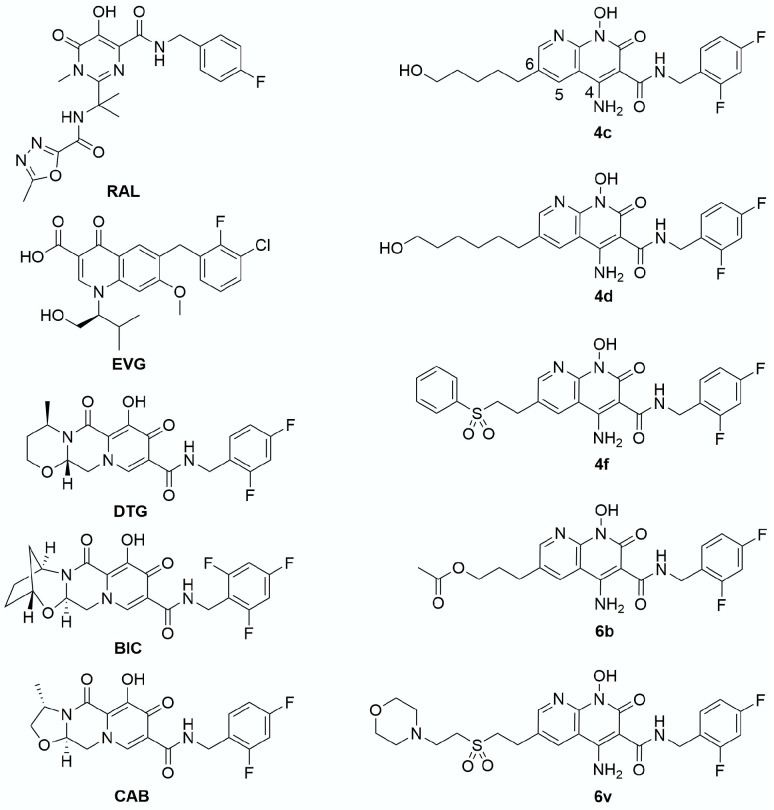
Chemical structures of the INSTIs. The chemical structures of the INSTIs used in this study are shown.

**Figure 2 viruses-13-02501-f002:**
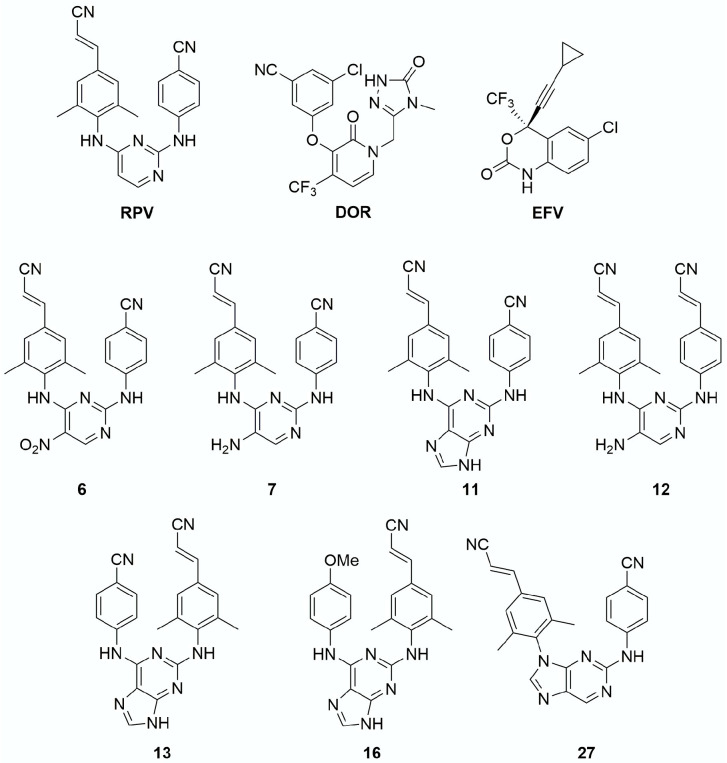
Chemical structures of the NNRTIs. The chemical structures of the NNRTIs used in this study are shown.

**Figure 3 viruses-13-02501-f003:**
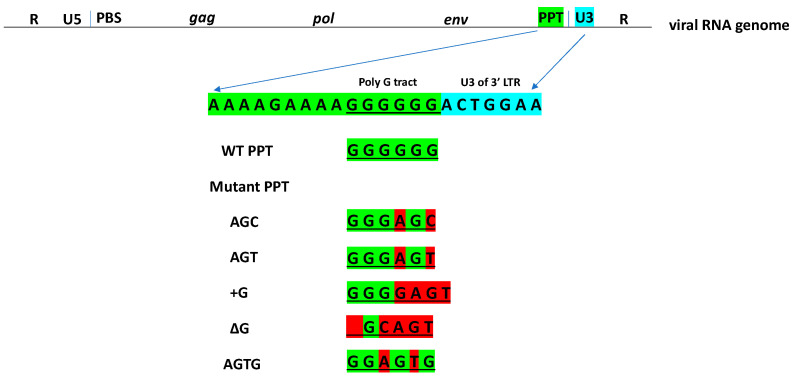
Design and construction of HIV-1 PPT mutants. Sequences of the HIV-1 PPT mutants. A schematic showing the mutations in the HIV-1 PPT. The HIV-1 PPT, which is highlighted in green, is adjacent to the U3 of the 3′ LTR (highlighted in cyan). The sequences of the HIV-1 PPT mutants are shown. The mutations, which are in in the poly G tract, are highlighted in red.

**Figure 4 viruses-13-02501-f004:**
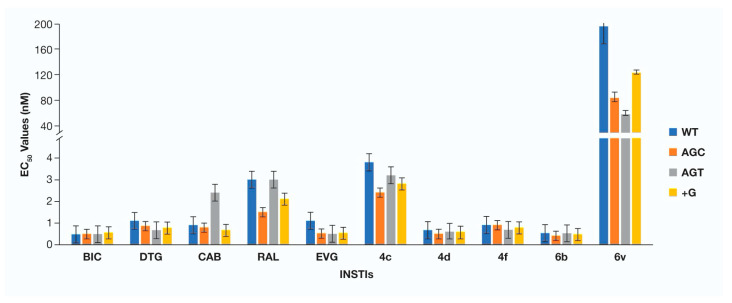
Antiviral potencies of INSTIs against a panel of HIV-1 PPT mutants. The EC_50_ values were determined using a vector that carries either a WT PPT or one of the HIV-1 PPT mutants in a single-round infection assay. The *Y*-axis, which depicts the EC_50_ value, is broken between 4 and 40 nM, and is set to a maximum of 200 nM. Error bars represent the standard deviation of the independent experiments, which were performed in triplicate.

**Figure 5 viruses-13-02501-f005:**
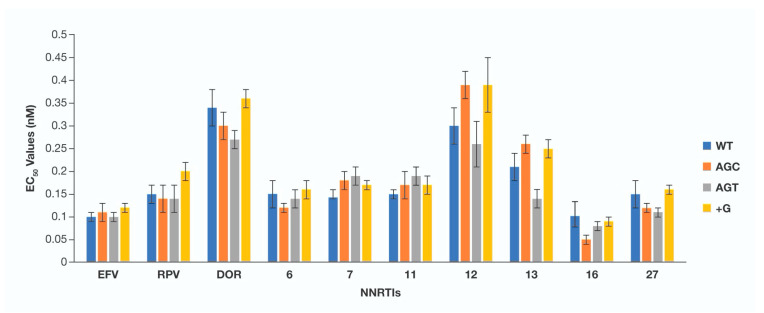
Antiviral potencies of NNRTIs against a panel of HIV-1 PPT mutants. The EC_50_ values were determined using a vector that carries either a WT PPT or one of the HIV-1 PPT mutants in a single-round infection assay. The *Y*-axis, which depicts the EC_50_ value, is set to have a maximum of 0.5 nM. Error bars represent the standard deviation of the independent experiments, which were performed in triplicate.

**Table 1 viruses-13-02501-t001:** Replication of the HIV-1 PPT mutants using a single round infection assay.

PPT Mutant	Single Round Infectivity (% of WT Activity)
AGC	5.0 ± 1.2
AGT	7.0 ± 1.4
+G	8.0 ± 2.5
ΔG	0 *
AGTG	0 *

* Single round infectivities for ΔG and AGTG could not be determined.

## Data Availability

The data presented in this study are available by request from the corresponding author.

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
