# Peer review of "INSTIs and NNRTIs Potently Inhibit HIV-1 Polypurine Tract Mutants in a Single Round Infection Assay"

_viruses, 2021, doi:10.3390/v13122501_

Round 1
Reviewer 1 Report
The authors present an important paper on HIV-1 drug resistance with relevance to clinical use of inhibitors. Most drug resistance to integrase strand transfer inhibitors (INSTIs) and non-nucleoside reverse transcriptase inhibitors (NNRTIs) map directly to the pol gene, either integrase or reverse transcriptase. It was shown previously that mutations that occur in the HIV-1 polypurine tract (PPT) gave rise to INSTIs and NNRTIs resistance while another report showed that drug resistance does not map to the PPT. The authors presence convincing evidence that PPT mutations do not give rise to INSTIs and NNRTIs drug resistance using single round infection assays. They clearly demonstrated that these drugs did not loss potency against wt virus and three characterized PPT mutants in single round infection assays. This report confirms that mutations that occur in the PPT do not give rise to drug resistance in either pol gene. They investigated clinical used drugs for both integrase and reverse transcriptase as well as other active drugs produced in their laboratory. The PPT mutants generally have a much decrease capacity to replicate relative to wild type mutants, i.e., around 5% of wt replication or are completely defective.
In summary, their data supports that HIV-1 PPT mutants themselves do not give rise to INSTI and NNRTI resistance.
P 3, line 70, of the, of the, is repeated
P5, lines 108, 109 have yellow highlight
P 8, line 207, extra space near end of sentence, also in line 217
P 8, third paragraph from end, Are there references that could be included here to support their reasoning, besides reference 41 in the preceding paragraph?
Author Response
The authors present an important paper on HIV-1 drug resistance with relevance to clinical use of inhibitors. Most drug resistance to integrase strand transfer inhibitors (INSTIs) and non-nucleoside reverse transcriptase inhibitors (NNRTIs) map directly to the pol gene, either integrase or reverse transcriptase. It was shown previously that mutations that occur in the HIV-1 polypurine tract (PPT) gave rise to INSTIs and NNRTIs resistance while another report showed that drug resistance does not map to the PPT. The authors presence convincing evidence that PPT mutations do not give rise to INSTIs and NNRTIs drug resistance using single round infection assays. They clearly demonstrated that these drugs did not loss potency against wt virus and three characterized PPT mutants in single round infection assays. This report confirms that mutations that occur in the PPT do not give rise to drug resistance in either pol gene. They investigated clinical used drugs for both integrase and reverse transcriptase as well as other active drugs produced in their laboratory. The PPT mutants generally have a much decrease capacity to replicate relative to wild type mutants, i.e., around 5% of wt replication or are completely defective.
In summary, their data supports that HIV-1 PPT mutants themselves do not give rise to INSTI and NNRTI resistance.
We appreciate the kind remarks and the positive review.
P 3, line 70, of the, of the, is repeated, Corrected.
P5, lines 108, 109 have yellow highlight, In the version we have, we do not see any yellow highlights. If we have missed something, please let us know.
P 8, line 207, extra space near end of sentence, also in line 217, Corrected.
P 8, third paragraph from end, Are there references that could be included here to support their reasoning, besides reference 41 in the preceding paragraph? Corrected, we included a reference.

Reviewer 2 Report
Smith, et al. perform a detailed exploration of the potency of antiretroviral (ARV) compounds, INSTIs and NNRTIs, against HIV-1 PPT tract mutants. The authors utilize both FDA-approved and candidate compounds to determine the ability of the HIV-1 PPT tract mutants to evade inhibition by ARVs. This study follows up on an observation that HIV-1 resistance mutations can arise outside of the target gene and may contribute to virological failure on INSTI- and/or NNRTI- containing regimens in vivo. However, recent observations suggest that the PPT tract mutants either are defective for HIV-1 replication in vitro or still retain efficacy against INSTIs and NNRTIs. Here, the authors expand upon those observations to measure the single-round infectivity of a panel of HIV-1 PPT tract mutants and calculated antiviral potencies of INSTIs and NNRTIs against WT and PPT tract mutants. The results presented here are in agreement with the observation that HIV-1 PPT tract mutations alone are not able to confer resistance to INSTIs and NNRTIs in vitro.
The manuscript is clear and of relevance to the HIV-1 drug resistance field. It is recommended that the authors address some minor comments and corrections to improve clarity and highlight significance for an audience that is less familiar with the field.
Major Comments
- Lines 81-84: It is recommended that the authors expand briefly on the “Cell-based assays” used to determine the single-cycle infectivity and EC50 This would be of a benefit to an audience that is unfamiliar with these standard assays in the HIV-1 field. Also, this would facilitate comparison of nuances between manuscripts describing similar phenotypes.
- Introduction; Lines 51-60: The Introduction would benefit from a brief summary of the findings of the paper and some brief conclusions. For example, content such as the paragraph in the “Conclusions” section (lines 169-177) would be informative for the audience.
- Conclusions: It is recommended that the authors comment on the disparity between the replication fitness of PPT mutant AGC reported in Wei, et al (ref 20) and the single-cycle infectivity results reported here.
- Figure 3: This figure may benefit from inclusion of nucleotide numbers for reference. For example, the Poly G tract spans from nt 9068 to nt 9073, depending on the reference sequence. Additionally, please define what is meant by “vir” on the right hand side of the HIV-1 genome.
- Table 1: The authors may want to consider indicating a lack of single round infectivity in the deltaG and AGTG virus as “ND” as opposed to “0.”
- Figure 4: Can the scale on the y-axis be adjusted to show the entire EC50 value measurement for viruses in the presence of INSTI 6v? This could be accomplished by inserting a break in the axis or by plotting 6v on a separate graph. This would also allow for the appreciation of the small fold change between WT and PPT mutant viruses in the presence of the majority of the INSTIs.
- Figure 5: Similar to comments on Figure 4, can the scale be adjusted to allow for the appreciation of the small fold change between WT and PPT mutant viruses in the presence of NNRTIs?
Minor Comments
- Lines 85-88: Please ensure that the compounds listed here in “INSTI and NNRTI synthesis” are the ones used for experiments in this manuscript. For example, compound 12 is shown in Figure 2, however, is not mentioned in this section. Similarly, compound 81 is mentioned in this section, but is not shown in Figure 2.
- Materials and Methods: Please include a section about how statistical analyses were carried out.
- Introduction: Please comment as to if any resistance mutants have been selected in vitro or in vivo in the presence of CAB.
- Please define the NNRTI “DOR” as it is not defined in the text.
- Line 255: Please clarify that the relative luciferase activity for the WT virus in the absence of ARVs was set to 100%.
Author Response
Smith, et al. perform a detailed exploration of the potency of antiretroviral (ARV) compounds, INSTIs and NNRTIs, against HIV-1 PPT tract mutants. The authors utilize both FDA-approved and candidate compounds to determine the ability of the HIV-1 PPT tract mutants to evade inhibition by ARVs. This study follows up on an observation that HIV-1 resistance mutations can arise outside of the target gene and may contribute to virological failure on INSTI- and/or NNRTI- containing regimens in vivo. However, recent observations suggest that the PPT tract mutants either are defective for HIV-1 replication in vitro or still retain efficacy against INSTIs and NNRTIs. Here, the authors expand upon those observations to measure the single-round infectivity of a panel of HIV-1 PPT tract mutants and calculated antiviral potencies of INSTIs and NNRTIs against WT and PPT tract mutants. The results presented here are in agreement with the observation that HIV-1 PPT tract mutations alone are not able to confer resistance to INSTIs and NNRTIs in vitro.
The manuscript is clear and of relevance to the HIV-1 drug resistance field. It is recommended that the authors address some minor comments and corrections to improve clarity and highlight significance for an audience that is less familiar with the field.
We appreciate the kind remarks and positive feedback.
Major Comments
- Lines 81-84: It is recommended that the authors expand briefly on the “Cell-based assays” used to determine the single-cycle infectivity and EC50 This would be of a benefit to an audience that is unfamiliar with these standard assays in the HIV-1 field. Also, this would facilitate comparison of nuances between manuscripts describing similar phenotypes. Corrected.
- Introduction; Lines 51-60: The Introduction would benefit from a brief summary of the findings of the paper and some brief conclusions. For example, content such as the paragraph in the “Conclusions” section (lines 169-177) would be informative for the audience. Corrected.
- Conclusions: It is recommended that the authors comment on the disparity between the replication fitness of PPT mutant AGC reported in Wei, et al (ref 20) and the single-cycle infectivity results reported here. We have made a comment in the revised manuscript about the fact that we cannot account for the difference in the data.
- Figure 3: This figure may benefit from inclusion of nucleotide numbers for reference. For example, the Poly G tract spans from nt 9068 to nt 9073, depending on the reference sequence. Additionally, please define what is meant by “vir” on the right hand side of the HIV-1 genome. Because of reporter gene has been deleted for env the numbering system would not correspond to HIV. We described our vector system in methods. We are not sure what the reviewer means by “vir”. It is possible that the reviewer has a truncated version of the figure (missing the right edge). The label at the right side of the drawing is “viral RNA genome” in the figure.
- Table 1: The authors may want to consider indicating a lack of single round infectivity in the deltaG and AGTG virus as “ND” as opposed to “0.” We think ND may be confusing; it could mean either “Not Done” or “None Detected”. We have amended the table to include a small explanation of what the “0” in the table means.
- Figure 4: Can the scale on the y-axis be adjusted to show the entire EC50 value measurement for viruses in the presence of INSTI 6v? This could be accomplished by inserting a break in the axis or by plotting 6v on a separate graph. This would also allow for the appreciation of the small fold change between WT and PPT mutant viruses in the presence of the majority of the INSTIs. Corrected.
- Figure 5: Similar to comments on Figure 4, can the scale be adjusted to allow for the appreciation of the small fold change between WT and PPT mutant viruses in the presence of NNRTIs? Corrected.
Minor Comments
- Lines 85-88: Please ensure that the compounds listed here in “INSTI and NNRTI synthesis” are the ones used for experiments in this manuscript. For example, compound 12 is shown in Figure 2, however, is not mentioned in this section. Similarly, compound 81 is mentioned in this section, but is not shown in Figure 2. Corrected.
- Materials and Methods: Please include a section about how statistical analyses were carried out. We did not carry out a statistical analysis of the comparisons of the data because there were no differences seen in the data. As noted in the figure legends, the drug susceptibility data are from triplicate assays, and the error bars show the standard deviation, so the reader can see how similar the data are.
- Introduction: Please comment as to if any resistance mutants have been selected in vitro or in vivo in the presence of CAB. Corrected.
- Please define the NNRTI “DOR” as it is not defined in the text. Corrected.
- Line 255: Please clarify that the relative luciferase activity for the WT virus in the absence of ARVs was set to 100%. Corrected, thanks.
We are grateful to the reviewers for their support, and helpful comments. We think that our manuscript has been substantially improved and hope that you find the revised the manuscript suitable for publication.
